



# The catastrophic thermokarst lake drainage events of 2018 in northwestern Alaska: Fast-forward into the future

Ingmar Nitze[1], Sarah Cooley[2], Claude Duguay[3, 4], Benjamin M. Jones [5], Guido Grosse[1, 6]
[1]Alfred Wegener Institute for Polar and Marine Research, Potsdam, 14473 Potsdam, Germany
[2]Department of Earth, Environmental and Planetary Sciences, Brown University, Providence, RI, 02912, USA
[3]Department of Geography and Environmental Management, University of Waterloo, Waterloo, Canada
[4]H2O Geomatics Inc., Waterloo, Canada
[5]Institute of Northern Engineering, University of Alaska Fairbanks, Fairbanks, Alaska, 99775, USA
[6]University of Potsdam, Institute of Geosciences, 14476 Potsdam, Germany
*Correspondence to*: Ingmar Nitze (ingmar.nitze@awi.de)
**Abstract.**
Northwestern Alaska has been highly affected by changing climatic patterns with new temperature and precipitation maxima
over the recent years. In particular, the Baldwin and northern Seward peninsulas are characterized by an abundance of
thermokarst lakes that are highly dynamic and prone to lake drainage, like many other regions at the southern margins of
continuous permafrost. We used Sentinel-1 synthetic aperture radar (SAR) and Planet CubeSat optical remote sensing data to
analyze recently observed widespread lake drainage. We then used synoptic weather data, climate model outputs and lake-ice
growth simulations to analyze potential drivers and future pathways of lake drainage in this region. Following the warmest
and wettest winter on record in 2017/2018, 192 lakes were identified to have completely or partially drained in early summer
2018, which exceeded the average drainage rate by a factor of ~10 and doubled the rates of the previous extreme lake drainage
years of 2005 and 2006. The combination of abundant rain- and snowfall and extremely warm mean annual air temperatures
(MAAT), close to 0° C, may have led to the destabilization of permafrost around the lake margins. Rapid snow melt and high
amounts of excess meltwater further promoted rapid lateral breaching at lake shores and consequently sudden drainage of
some of the largest lakes of the study region that likely persisted for millenia. We hypothesize that permafrost destabilization
and lake drainage will accelerate and become the dominant drivers of landscape change in this region. Recent MAAT are
already within the range of predictions by UAF SNAP ensemble climate predictions in scenario RCP6.0 for 2100. With MAAT
in 2019 exceeding 0° C at the nearby Kotzebue, Alaska climate station for the first time since continuous recording started in
1949, permafrost aggradation in drained lake basins will become less likely after drainage, strongly decreasing the potential
for freeze-locking carbon sequestered in lake sediments, signifying a prominent regime shift in ice-rich permafrost lowland
regions.



Keywords: Permafrost, permafrost thaw, thermokarst, lake change, lake drainage, Seward Peninsula, Baldwin Peninsula, Alaska

## 1 Introduction

Permafrost is widespread (20 to 25 % of the land area) in the northern high latitudes (Brown et al., 1997; Obu et al., 2019) and is primarily a result of past and present cold climatic conditions (Shur and Jorgenson, 2007). The rapidly warming Arctic climate is already reducing the stability and distribution of near-surface permafrost. Warming of permafrost at the global scale has been observed over recent decades from borehole temperature measurements (Romanovsky et al., 2010; Biskaborn et al., 2019), while local to regional permafrost degradation has been observed in many studies of varying scales across the permafrost domain (Nitze et al. 2018). Widespread near-surface permafrost loss or transition from continuous to discontinuous permafrost has for example been shown with remote sensing-supported permafrost modeling in Alaska (Pastick et al., 2015). Permafrost degradation may lead to long-term surface subsidence (Streletskiy et al., 2017), change in hydrological regimes (Liljedahl et al., 2015), and release of greenhouse gases carbon dioxide ($CO_2$), methane ($CH_4$), or nitrous oxide ($N_2O$) (Elberling et al., 2013; Walter Anthony et al., 2018; Repo et al., 2009). In particular, the release of greenhouse gases from carbon locked away for thousands of years will trigger further warming through the permafrost carbon feedback (Schuur et al., 2015). Furthermore, the stability of permafrost is crucial for local communities which are dependent on ground stability for infrastructure, food security, and water supply (Chambers et al., 2007; White et al., 2007; Melvin et al., 2017; Hjort et al., 2018).

Rapid changes in lake area, including expansion and drainage, are strong indicators of permafrost degradation and thaw (Smith et al., 2005; Hinkel et al., 2007; Jones et al., 2011; Grosse et al., 2013; Arp et al., 2018; Nitze et al., 2018a). Natural lake drainage has been associated with near-surface permafrost degradation such as melting of ice wedges, formation of thermo-erosional channels, gully headward erosion, or internal drainage through permafrost-penetrating taliks (Mackay, 1988; Yoshikawa and Hinzman, 2003; Hinkel et al., 2007; Marsh et al., 2009; Jones et al., 2020a). Other natural lake drainage events have been connected to increased precipitation, causing bank overtopping with subsequent drainage channel formation, or snow dams and subsequent outburst floods and drainage channel formation (Mackay, 1988; Jones and Arp, 2015).

The southern margin of continuous permafrost in northwestern, western and interior Alaska as well as adjacent northwestern Canada has been identified as a region with a high temporal variability in lake area, and particularly widespread lake drainage (Jones et al., 2011; Chen et al., 2014; Lantz and Turner, 2015; Nitze et al., 2018a). Over the past few decades, lake drainage has outpaced lake growth by 14.9 % on the Seward Peninsula in western Alaska, largely driven by the drainage of several very large individual lakes (Jones et al., 2011; Nitze et al., 2017). Other transitional permafrost regions around the Arctic are broadly affected by the same pattern, with widespread drainage events and total area of lake loss exceeding total area of lake expansion (Smith et al., 2005; Nitze et al., 2018a). However, lakes also experience intra-annual (Cooley et al., 2017, 2019) to multi-year (Plug et al., 2008; Karlsson et al., 2014) water level fluctuations linked to precipitation and evaporation dynamics or overall hydrological runoff regimes, which can cause high uncertainty in interpreting temporally sparse observations. In particular,



recently shifting weather patterns with warmer air and sea surface temperatures along Arctic coasts driven by reduced sea ice
cover (Bhatt et al., 2014) may also have an effect on coastal lowland permafrost (Lawrence  and Slater, 2005) and thus
potentially lake dynamics (Alexeev et al., 2016; Arp et al., 2019). For example, persistent warm air and sea surface
temperatures caused a new sea ice minimum in the Bering Sea west of Alaska resulting in unprecedented largely open seas in
the winter 2017/2018 (Stabeno and Bell, 2019) .
Other regions with cold continuous permafrost (e.g., Arctic Coastal Plain, Tuktoyaktuk Peninsula, Coastal Lowlands of
Siberia) are also affected by lake drainage (Hinkel et al., 2007; Kravtsova  and Bystrova, 2009; Karlsson et al., 2012; Lantz
and Turner, 2015; Olthof et al., 2015; Nitze et al., 2017, 2018; Jones et al., 2020a), but to a lesser intensity than the transitional
zone towards discontinuous permafrost, particularly in Alaska and western Siberia (Nitze et al., 2017, 2018).
Several studies suggest that lake drainage might be episodic with drainage events clustered in time and therefore potentially
related to specific environmental conditions, such as high precipitation events (Marsh et al., 2009; Swanson, 2019; Jones et
al., 2020a). Others, in contrast, find more stable to decreasing, long-term drainage rates, e.g. in northern Alaska and the Western
Canadian Arctic (Hinkel et al., 2007; Marsh et al., 2009;  Jones et al., 2020a).
In western Alaska, a series of major drainage events took place in the mid-2000s, where some of the largest thermokarst lakes
on the ground-ice rich northern Seward Peninsula drained within a short period of a few years (Jones et al., 2011; Swanson,
2019). The recent drainage of several large lakes on the northern Seward Peninsula and the largest lake on the Baldwin
Peninsula provides an interesting test bed for analyzing lake drainage progression in high temporal and spatial detail using
remote sensing imagery, meteorological data, and lake ice characteristics. The geographic proximity to the Bering and Chukchi
seas that both have experienced rapid sea ice loss and climatic shifts in recent years offers a unique opportunity to study the
relationship between changing climate regimes and lake dynamics in permafrost regions on short-time-scales. In this study we
therefore use temporally high-resolution remote sensing and meteorological data to quantify:
1)   How much lake area was affected by the recent drainage events in western Alaska in 2018?
2)   How do the drainage events, documented in 2018, compare to other previous events such as in the mid 2000's in
86            terms of area, spatial distribution, and temporal (intra-annual) sequence?

3)   What are the primary drivers of the recent drainage events and how may projected future climate scenarios affect lake
88            trajectories in this region?

To answer these questions, we analyzed recent optical and synthetic aperture radar (SAR) satellite imagery (Planet, Sentinel-
1) from 2017 and 2018 to map the spatio-temporal lake change dynamics and compared the results to available datasets of past
lake dynamics (Nitze et al., 2018a; Nitze et al., 2018b) and climatic conditions. Furthermore,  we investigated weather and
climate data as well as modeled lake ice conditions as potential drivers of the widespread lake drainage.





## 2 Study area

In this study, we focus on the northern Seward Peninsula (NSP) and the Baldwin Peninsula (BP) in western Alaska. The study area covers a total land area (including interior water bodies) of 25,271 km². It is bounded by the Chukchi Sea and Kotzebue Sound to the north and northwest, different hill ranges in the south, and Selawik Lake, Hotham Inlet and the 161°W meridian in the east (Figure 1). It is part of the Bering Land Bridge region, which was largely unglaciated during the last glacial maximum and is now located at the southern margin of the continuous permafrost zone (Jorgenson et al., 2008; Obu et al., 2019). Measured ground temperatures range between -3.5 and –0.8 °C (Biskaborn et al., 2015; GTN-P Ground Temperature Database). Modeled ground temperatures range between -2.8 and +0.5 °C, with the majority between -1.5 and -2.0°C (Obu et al., 2019).

The area is characterized by a subarctic continental climate with a mean annual air temperature (MAAT) of -5.1 °C and 279 mm precipitation as reported at the Kotzebue climate station (NOAA, 1981-2010). Snowfall accumulation averages 157 cm per year, considerably more than for example in northern Alaska (~95 cm in Utkiagvik/Barrow). Snow typically persists until the mid to end of May (Macander et al., 2015).

The study region is composed of a strongly degraded ice-rich permafrost landscape with typical permafrost landforms, such as thermokarst lakes and drained thermokarst lake basins of several generations (Plug and West, 2009; Jones et al., 2011; Jones et al., 2012), pingos, ice wedge polygon networks, and ice-rich yedoma uplands as remnants of the Pleistocene accumulation plain (Hopkins et al., 1955; Jongejans et al., 2018). The morphology is variable with mostly flat terrain (<20 m) in highly degraded permafrost terrain along the coastal margin of the NSP and undulating terrain with steep slopes in the upland regions of the NSP. The mountainous terrain along the southern margin of the NSP reaches up to ~700 m elevation. The Baldwin Peninsula (BP) is characterized by rolling terrain from sea-level to ~50 m elevation with a mixture of degraded permafrost with partially drained lake basins and uplands in various stages of degradation (see Figure 1).

The foothills and mountain ranges of the study area are underlain by bedrock. Furthermore, the NSP is locally affected by Late Quaternary volcanism, with the presence of four known maar lakes (Devil Mountain, White Fish, North Killeak, South Killeak), which are the largest lakes of the study region and the largest maar lakes globally (Beget et al., 1996). Further volcanic landscape features such as degraded volcanic bedrock cores, young basaltic lava flows and young cinder cones are locally present in the southern portion of the NSP (Hopkins, 1955). In part, deposits of the BP are likely of glacial origin, with a buried terminal moraine covered with yedoma-like, ice-rich sediments (Huston et al., 1990; Jongejans et al., 2018).

The region is one of the major lake districts in Alaska (Arp and Jones, 2009). Lake presence in the selected study area is concentrated on the coastal plains and thermokarst terrain. The majority of lakes are located in drained thermokarst lake basins, have shallow depths of less than 2 m, and are often later generation thermokarst lakes in locations that experienced several previous lake generations (Jones et al., 2012; Lenz et al., 2016). However, first generation thermokarst lakes up to 15 m in depth, intersecting the remaining yedoma upland surfaces, are still present (Kessler et al., 2012). Yedoma uplands with flat





surfaces are speckled with initial thermokarst ponds and small lakes, most notably on the BP (Jongejans et al., 2018). In
addition, the four large maar lakes on the NSP that reach depths of up to 100 m (Beget et al., 1996).
Vegetation is predominantly composed of shrubby tundra and is located in zones D and E of the Circumpolar Arctic Vegetation
Map (CAVM) (Walker et al., 2005). Vegetation is typically abundant in sheltered areas along thermokarst lake margins.
Floating vegetation mats may be present on lakes margins and persist above water associated with expanding lake margins
(Parsekian et al., 2011).

## 131 3 Data and Methods

### 132 3.1 Data

#### 133 3.1.1 Lake dataset base layer

We used the lake change dataset of Nitze et al. (2018a, 2018b) as the base layer for further analysis. This dataset contains
polygon vectors of the buffered lake extent of individual lakes larger than 1 ha. It includes spatial attributes and statistics such
as individual lake area in 1999 and 2014, net change (gain minus loss) and gross changes (gain, loss) from 1999-2014, as well
as lake shape parameters, such as orientation, eccentricity, and solidity. All lakes intersecting the study area (n=4605) were
selected for analysis. Further GIS and spatial analyses are based on the geometries of this lake dataset (Lake Change 1999-
2014: named Lk hereafter). An overview of the lake change datasets is provided in Table 1.

### 140 3.2 Remote sensing analysis

#### 141 3.2.1 Water masks for 2017 and 2018: Sentinel-1 imagery

We extracted late-summer water masks for the years 2017 and 2018 using Sentinel-1A/B SAR data in Google Earth Engine
(GEE) (Gorelick, et al., 2017) (see Figure 2). We identified all Sentinel-1A/B images available between 1 August and 30
September in both 2017 and 2018 (Watermask 2017: WM2017, Watermask 2018: WM2018) and selected VV polarization,
which was available for all S1-data within this period. Erroneous low-backscatter values along image margins, which are a
common issue for these datasets, were clipped per default with a buffer of 5000 m.
We created a median value composite of the entire image stack, to lessen the impact of very high backscatter values caused by
windy conditions. After histogram analysis, we determined a backscatter value of -18 dB as the best threshold point between
land and water. All backscatter values below -18 dB were added to the surface water masks of 2017 (WM2017) and 2018
(WM2018), respectively. We exported the two water masks to raster files with 20x20 m grid spacing in UTM3N projection.
The water masks WM2017 and WM2018 were intersected with the lake extent base layer (Lk) using zonal statistics in QGIS
version 3.6 (QGIS Development Team, 2019) to retrieve lake area extent and zonal statistics values for 2017 and 2018.



Lastly, all lakes with a lake area loss of >25 % and initial size of >1 ha, based on the difference of WM2017 and WM2018,
were defined as drained lakes. This follows previous studies defining drainage thresholds by lost water area of >25 % (Hinkel
et al., 2007; Olthof et al., 2015; Jones et al., 2020a). The drained lakes dataset is referred to as LkDrain.

Links to the GEE code used for water masking are provided in the Code and Datasets Section.

**3.2.2 Timing of drainage 2017 and 2018: Planet imagery**

To determine the drainage patterns and mechanics as well as to compare long-term versus short-term drainage patterns, we
automatically analyzed temporally high-resolution Planet CubeSat imagery (Planet Team, 2017) from 2017 and 2018 and
visually inspected the largest drained lakes. With over 120 satellites in orbit, the Planet constellation provides a temporal
frequency of observations of less than one day at a ground resolution of 3.125 m, which makes Planet data an ideal solution
for mapping rapid landscape dynamics at high spatial and temporal resolutions. For mapping individual lake dynamics in 2017
and 2018, we used the automated lake tracking workflow presented in Cooley et al. (2017, 2019). A complete description of
the method can be found in Cooley et al. (2019). A brief summary is provided here. First, we downloaded all PlanetScope
(3.125 m resolution) and RapidEye OrthoTiles (5 m resolution) with <20 % cloud cover available from Planet Labs between
May 1 and October 1 for both 2017 and 2018. We then created an initial lake mask which contains the maximum extent of all
water bodies in the study area between 2017 and 2018. This initial mask was buffered by 60 m and all rivers were removed to
produce a buffered water mask used for both seeding the water classification and tracking changes in lake area.
We then classified all of the images into water or land by applying a histogram-derived threshold to each image's NDWI
((NIR–green) / (NIR+green)) as described in Cooley et al. (2017; 2019). To track changes in lake area, we used an object-
based lake tracking method wherein for every image, we calculated the total amount of water contained within each lake object
in the buffered mask. This method allows for direct comparison between RapidEye and PlanetScope imagery with its different
spatial resolution and furthermore is robust against potential minor geolocation uncertainty.
At the time of analysis, Planet Labs imagery did not provide a reliable cloud mask. Therefore, the third and most critical step
of the method was removal of cloudy or poor quality observations using a machine learning-derived filtering algorithm. To do
this, we first created a manual training dataset of valid/invalid lake area observations and then used this dataset to build a
random forest classifier that automatically removes cloudy/poor quality lake observations. This method is able to accurately
classify 97 % of observations as valid or invalid. We then selected the best observation for each day and applied additional
outlier and median filters to produce the final time series. While we do not specifically remove ice-covered observations from
the analysis, Cooley et al. (2019) demonstrate that most ice-impacted lake area observations are classified as invalid by the
random forest classifier.



The final lake dynamics dataset, henceforth referred to as LkDyn, includes buffered polygon vectors, seasonal time series of
lake area, as well as basic descriptive lake area statistics such as minimum area, maximum area, and seasonal dynamics (max
- min) for each individual lake. For the analysis of temporal lake drainage patterns we spatially joined all lakes of LkDyn,
which intersected LkDrain.

### 3.2.3 Identifying past lake drainage for 1999-2014

For lake dynamics from 1999-2014, we used the lake change dataset of Nitze et al. (2018) (Lk) to compare recent dynamics
to the observed drainage events of 2018. We opted for manual image interpretation based on satellite imagery video animations
as there is to our best knowledge no reliable automated method available to determine drainage dates in challenging Arctic
environments. We tested the automated LandTrendr method, which automatically determines breakpoints in time-series, to
retrieve the timing of lake drainage between 1999 and 2018 (Kennedy et al., 2010; Kennedy et al., 2018). Results obtained
with this method were highly unstable with insufficient reliability.
We created video animations in GEE for each individual drained lake, with time-stamped frames, and determined the drainage
year manually through visual interpretation (link to code see below). The drainage year was defined as the point in time of
initial clearly visible drainage, which could be a) visible exposure of lake bottom sediments or b) a strong increase in
vegetation, e.g. due to sudden lake level drop. The entire calculated area loss was assigned to the determined drainage year.
Lake area loss of lakes with a longer drainage process >1 year, e.g. from 2005 until 2009, were counted as full drainage in the
initial drainage year (2005). The visual interpretation was aided by plotting the time-series of multi-spectral indices (Tasselled
Cap, NDVI, NDWI) for each individual drained lake.
Lakes with data gaps (up to several years) right before the determined drainage year, were flagged in the statistics. This
frequently applied to years 2005 and 2008, which had several data gaps in the preceding years (see Supplementary Figure 1).
Data gaps were caused by limited data availability, frequent cloud cover and shadows, as well as the Landsat-7 Scan Line
Corrector (SLC) error. Lakes where the timing could not be detected manually, e.g. in case of very subtle drainage, were
assigned no drainage year (25 of 270).
Links to the GEE video animation processing code and time-series plotting are provided in the Code and Datasets Section.
The videos are accessible at:
[https://github.com/initze/NW_Alaska_Drainage_Paper/tree/master/animations/lake_animations_drainage_1999-2014](https://github.com/initze/NW_Alaska_Drainage_Paper/tree/master/animations/lake_animations_drainage_1999-2014)

### 3.3 Climate and weather analysis

### 3.3.1 Weather

We analyzed synoptic weather data from the nearest weather station in Kotzebue that is provided by the National Oceanic and
Atmospheric Administration (NOAA). We acquired the GHCN-Daily datasets (Menne et al., 2012) in CSV format through



the web-search on the NOAA website (https://www.ncdc.noaa.gov/cdo-web/search). The dataset provides a daily series of
minimum ($t_{min}$) and maximum ($t_{max}$) temperatures (t), precipitation (prcp) and snowfall (sf) from 1897 until 2019, with
continuous observations since 1950. Daily mean temperatures are not available continuously. Therefore, we calculated daily
mean temperatures as the mean of daily minimum and daily maximum temperatures. For calculating the influence of winter
conditions (t, prcp, sf) we analyzed the weather conditions from July 1 of the preceding year until June 30 on a yearly basis,
from here on referred to as "winter year". Therefore, winter year 2018 for example is defined as the period from July 1 2017
through June 30 2018. In addition to the standard attributes (mentioned above), we calculated Freezing Degree Days (FDD)
as the cumulative sum of negative mean daily temperatures per winter year. Snow accumulation is calculated as the cumulative
sum of snowfall per winter year.
We calculated climatological means for daily observations and yearly aggregated statistics. For daily values we calculated
means and standard deviations of mean temperatures ($t_{mean}$) for each calendar day, excluding 29 February, from 1 January 1981
through 31 December 2010. We calculated yearly mean temperature as the mean of daily $t_{mean}$ mean temperatures ($t_{mean}$). We
calculated the mean of yearly values between 1981 and 2010 as the climatic mean temperature. For annual statistics of winter
years we calculated values ranging from July 1980 through June 2010, according to the previously stated winter year definition.
Code: For climate and weather data preprocessing and time-series plotting, a python package was developed by Ingmar Nitze,
which is available at https://github.com/initze/noaaplotter.
**3.3.2 Climate prediction**
We downloaded Decadal SNAP (Scenarios Network for Alaska and Arctic Planning, 2020) Ensemble Climate Model
Projections (2 km CMIP/AR5) of Scenarios RCP4.5, RCP6.0, and RCP8.5 for the study region. This dataset contains decadal
(2000-2010, 2010-2020, …, 2090-2100) mean annual, seasonal and monthly air temperature and precipitation. For analysis
we used annual predictions of temperature (MAAT) and precipitation (MAP). Gridded data is available at a spatial resolution
of 2 km across Alaska and parts of western Canada. We clipped the data to the extent of the study area and calculated the mean
and standard deviations for the entire study region for projected MAAT and MAP values for each decade.
**3.4 Lake ice simulations**
We used the Canadian Lake Ice Model (CLIMo; Duguay et al., 2003) to analyze the impact of weather conditions on lake ice
growth and permafrost. CLIMo is a 1-D thermodynamic ice model that has been used in several studies (Ménard et al., 2002;
Labrecque et al., 2009; Brown and Duguay, 2011; Surdu et al., 2014; Antonova et al., 2016). CLIMo output includes all energy
balance components, on-ice snow depth, the temperature profile at an arbitrary (specified) number of levels within the ice/snow
(or the water temperature if there is no ice) and ice thickness (clear ice and snow ice) on a daily basis, as well as freeze-
up/break-up dates and end-of-season clear (congelation) ice, snow ice and total ice thickness. Model output of particular
interest to lake ice simulations within the context of this study is the evolution of lake ice growth and maximum ice thickness



as they are useful proxies for freezing intensity and the influence of weather conditions on potential ground stability.
Thicknesses of snow ice and that of congelation ice layers (referred to hereafter as "top-growth" and "bottom-growth",
respectively) were also analyzed to account for snow mass and snow insulation effects.
The lake ice model is forced with five meteorological variables consisting of mean daily near-surface air temperature, relative
humidity, wind speed, cloud cover, and snowfall (or snow depth from a nearby land site when available). Four of the five
meteorological variables (all but snowfall) were taken directly or derived from the ERA5 atmospheric reanalysis product from
the European Centre for Medium-Range Weather Forecasts (ECMWF). Since ERA5 did not provide adequate snowfall or
snow depth values, we obtained snow depth data from NOAA's Global Historical Climate Network Daily (from the nearest
weather station at Kotzebue Ralph Wien Memorial Airport) for model simulations.
We performed simulations over nearly a 40-year period (1980-2018) and with specification of a mixed-layer depth of 2 m.
The length of the record was chosen based on the availability of ERA5 data and to be able to place lake ice model output for
the 2018 winter year into a broader historical context. Finally, in order to account for redistribution of snow across lake ice
surfaces which is a process well documented in several studies (e.g. Duguay et al., 2003; Sturm and Liston, 2003; Brown and
Duguay, 2011; Kheyrollah Pour et al., 2012), we ran the model with two sets of snow depth scenarios; one with full snow
cover (100% of the amount measured at the Kotzebue weather station) and the other with no snow cover (0% snow – i.e. snow
free ice surface) to capture the range of snow conditions that one would expect to observe in the field.
**4 Results**
**4.1 Lake changes**
**4.1.1 Lake drainage 2018**
Lake area loss was severe in 2018, where 192 of 4605 lakes larger than 1 ha lost more than 25 % of their initial size (LkDrain).
These lakes lost an accumulated water area of 1622.04 ha between late summer 2017 and 2018. Total net lake area loss,
including all lakes, was 2062.56 ha (4 % of the total lake area in the study domain).
Lake drainage clustered around two types of lake sizes. Five very large lakes (>100 ha) lost 1072.68 or 66.1 % of the total
drained lake area (LkDrain), while the remaining 190 lakes accounted for 549.36 ha or 34.9 %, where the largest lake had an
initial size of 28.5 ha in 2017 (Table 2). Of the five large drained lakes, four are of thermokarst origin and the largest is a
lagoon on the BP, which likely is affected by episodic flooding and drying. The five large lakes that drained were some of the
largest thermokarst lakes in the entire study area before drainage (Size rank 6, 12, 32, 39, 51). The only other bigger lakes in
the study region were formed or affected by Late Quaternary volcanic activity (four maar lakes and Imuruk Lake) and therefore
are less prone to lake drainage caused by permafrost degradation.
Spatially, the highest density of lake drainage events is located in the Cape Espenberg region in the northeastern part of the
NSP (see Figure 3). On the BP, two spatial clusters of lake drainage prevail. The first cluster is located in the center of the





northern part of the BP, which encompasses the now drained formerly largest lake (Lake ID 64656) and its neighboring basins.
The second cluster is located in the southern part of the BP, where several partially drained lakes form a nearly linear structure.
Smaller clusters or individual lake drainage events are scattered predominantly along coastal and lowland areas of the entire
study region and across different landscape units, such as uplands, thermokarst basins, coastal depressions or river floodplains.
Lake drainage in the southern more mountainous region of the SP was scarce.
**4.1.2 Intra-annual lake drainage dynamics**
**Temporal Patterns**
The analyzed lakes exhibit various distinct seasonal patterns of lake area loss or drainage. The ice-break-up period in late May
and early June 2018 was the most dominant period of lake drainage. Nine of the largest 10 lakes (see Table 3) exhibit a strong
decline in lake area before July 2018, and one rapid drainage event in early July (Lake ID 101359). In the majority of these
cases (n=8), the first valid observation of 2018 already shows a significant decline compared to the last observation of 2017,
which indicates drainage during snow-melt and ice-break-up, when data observations were still masked due to the presence of
ice and snow. During June 2018 weather conditions were favorable for optical remote sensing and observations for ice-free
persistent lakes are available. From July lake area only decreased slowly and gradually among the analyzed lakes (LkDrain)
without further distinct drainage peaks. A detailed example of a representative lake drainage event is presented in Figure 4.
Apart from the general regional dynamics, individual lake drainages followed variable patterns of drainage velocity/duration
and timing. Drainage patterns included sudden complete lake area loss (e.g. Lake IDs 99230, 64656), multiple recurring
drainage events (Lake IDs 72420, 100644, 99583), gradual loss (Lake IDs 99756, 100218) to initial loss followed by partial
refilling (Lake IDs 99381, 99465, 99532) (see Supplementary Files).
Supplementary figures are available at:
https://github.com/initze/NW_Alaska_Drainage_Paper/tree/master/figures/lake_drainage/planet_lake_area

**Quantification**
The early season lake drainage affected the largest lakes, and therefore the largest area. The time-series animation of the ten
largest lakes can be accessed by video (see Table 2). The third largest drained lake (Lake ID 99230) for example started
draining on June 2 and lost the majority of its water within the following two weeks. During the summer months the remaining
shallow ponds dried out further, while only few apparently deeper ponds remained. Imagery from spring 2019 showed the
development of vegetation, which follows the typical thermokarst lake cycle of this region (lake, lake drainage, drying of
exposed lake bottom, vegetation emergence; Jones et al., 2012) (see Table 2 for video). The largest drained lakes follow a
similar trajectory of rapid drainage around ice-break-up and further drying of the drained lake basin during peak summer.





**Spatial patterns**

Nine out of the ten largest drained lakes are second generation thermokarst lakes, which are typically located within a complex of previously drained lake basins. The second largest "lake" is actually a lagoon, which is likely influenced by sea water inundation. Each of the lakes had a significant fraction of their shoreline within the former drained lake basin. These are typically covered by wet tundra and underlain by terrestrial peat overlying lacustrine sediments (Jones et al., 2012). Based on visual image interpretation, all lakes drained through previously established drainage pathways, which are located in flat basin terrain and suggest that these likely are "weak spots" for full drainage.

During these drainage events new channels formed or existing channels deepened. In several instances (Lake IDs 99368, 64656, 99492, 102499), new drainage channels are evident based on new sediment fans that formed downstream. In the case of several lakes, the drainage caused a chain reaction, where hydrologically connected lakes, both up- and/or downstream of the initially drained lake, drained as well. Due to the widespread presence of these surface drainage indicators, talik penetration to a groundwater layer can be excluded for the study area.

**4.1.3 Lake drainage 1999-2014**

From 1999 through 2014 we observed 268 lakes larger than 1 ha that lost more than 25 % of their area, resulting in a total water area loss of 3245.74 ha during the observed period within this group of lakes. The net lake loss of the study region, including all lakes was 3677.43 ha or 6.0 % of the overall lake area. The six largest drained lakes accounted for more than half of the lost lake area (50.6 %) and were each among the 33 largest lakes (Size rank 9, 12, 13, 26, 29, 33) of the study area (Table 3). Each of these lakes was apparently of thermokarst origin.

These drained and partially drained lakes predominantly occur along the near coastal zone of the NSP and around Shishmaref Inlet (see Figure 3). Within this region, lake drainages are distributed uniformly with no distinct clusters. The BP and southern Kotzebue Sound do not show major drainage activity in this period with only three lakes that fulfill the defined criteria. The vicinity of Imuruk Lake had four drained lakes.

**Timing of drainage**

The analysis of drained lakes revealed a period of widespread lake drainage with up to 21 confirmed events per year from 2002 until 2007 and 2009 (see Supplementary Figure 2). The number of detected drained lakes in 2005 was exceptionally high with 56, but the majority (n=33) did not have sufficient observations in the preceding year 2004 or even 2003 to confirm the correct drainage year. This number of drained lakes is therefore associated with a high degree of uncertainty. Years 2003, 2007, 2009 and 2012 also have more than 5 lakes, which have an uncertain drainage date.

Although the number of drained lakes is relatively stable over time, drained lake area spiked in 2006 and 2007 with 922 ha (uncertain: 6.35 ha) and 631 ha (uncertain: 15.57 ha) net lake area loss, respectively. The significant uptick in 2006 was driven





by the drainage of very large lakes, particularly in the Cape Espenberg region of the NSP. The years 2003 and 2004 follow
with lake area loss of 323.65 ha (uncertain: 41.58 ha) and 413.06 ha (uncertain: 3.91 ha), respectively. The numbers are
conservative and might be even higher (see uncertainty 2005) due to a low data coverage during this period.
**4.2 Weather and Climate**
**4.2.1 Weather observations**
The preceding winter and spring of 2017/2018 (winter year 2018) was the warmest, wettest and second snowiest on record at
this time. Compared to the entire weather record this winter was highly exceptional (Figure 5, Table 4). Mean daily air
temperatures exceeded the climatological means persistently and frequently with 127 days above one standard deviation from
the climatological mean, only interrupted by a short cold snap in January 2018 (Figure 6). On several days air temperatures
were close to 0 °C, which is 15 to 20 °C above the climatological mean. Exceptional warmth lasted continuously from October
2017 until fall 2019. Winter year 2019 even surpassed 2018 with an annual air temperature of +0.12 °C, but with average
precipitation and snowfall accumulation of 279 mm and 155 cm, respectively.
The weather station recorded only 1905 cumulative Freezing Degree Days (FDD; the sum of average daily degrees below 0
°C) and an annual air temperature of -1.3 °C, which exceeded the previous record by 0.53 °C and 238 FDD. The 10 warmest
and 5 coldest years are shown in Table 5. Accumulated snowfall was the 2nd highest on record with 274 cm, only exceeded by
2005 (305 cm). Overall precipitation of the winter year 2018 was the highest on record with 424.5 mm, exceeding very wet,
but much colder winter years 2013 (402 mm, -5.41 °C) and 1995 (393.7, -5.92°C). Precipitation, mostly as snowfall, was
particularly strong from October through February, with the exception of January.
All indicators highlight the exceptional conditions of winter 2017/2018 in western Alaska. Weather data from Nome (ca. 300
km south of Kotzebue) on the southern SP indicate a similar picture of extreme weather conditions with the second warmest
and third-snowiest winter year on record. Climate reanalysis data (GHCNv4) confirm a larger regional pattern of exceptionally
warm conditions across the Bering Strait (see Supplementary Figure 2).
**4.2.2 Climate model projections**
The UAF SNAP Climate model ensemble consistently projects an increase in temperature and precipitation for western Alaska
with a plateau after around 2070 for RCP4.5 and continuous increase for the remaining scenarios in the 21st century. They
predict an increase to a regional MAAT of -0.39 ± 0.38°C (RCP4.5), +0.44 ± 0.37 °C (RCP6.0), and +3.00 ± 0.38 °C (RCP8.5)
during the 2090s, which marks an increase of 3.7 to 6.6 °C (Supplementary Figure 3). MAP is projected to increase by around
12 % (RCP4.5), 20 % (RCP6.0), and 32 % (RCP8.5) on average.



**4.3 Lake ice simulations**

Modeled maximum lake ice thickness of the winter year 2018 was 1.14 (100% snow) to 1.32 m (no snow). It was below average compared to 1981-2017 (1.31 m ± 0.14 m for 100% snow; 1.68 m ± 0.12 m for no snow) but thicker than the absolute minimum of 0.99 m (100% snow) in 2014. Lake ice thickness of 2018 was primarily determined by top ice-growth (snow-ice formation), which is strongly dependent on snow mass on the ice surface. Snow-ice formation correlates well with high snowfall years, such as 2005 or 2011. The bottom (congelation) ice-growth of 2018 reached a new extreme low with only 0.32 m (100% snow) to 1.32 m (no snow) (see Figure 7). This compares to 0.87 m ± 0.23 m (100% snow) and 1.68 m ± 0.12 m (no snow) from 1981-2017. Low bottom ice-growth indicates a strongly decreased freezing activity (negative heat flux) into the lake and potentially into the ground of the surrounding terrain. The exceptional combination of high temperatures and high snowfall, as experienced in ice season 2017-2018, are the strongest factors for these patterns.

**5 Discussion**

**Lake drainage in western Alaska in historical context**

The massive drainage of many lakes in early summer 2018 in western Alaska was an extreme event, which dwarfs previous lake drainage events within this region since the availability of remote sensing data. Although the study area experienced widespread lake drainage during the mid 2000's (Jones et al., 2011, Nitze et al., 2018a; Swanson, 2019), the year 2018 exceeded average annual lake drainage rates of 1999-2014 by a factor of 7.5 in area and 10.9 in numbers of lakes, clearly indicating a response of the system to extreme weather conditions. Recent lake drainage in 2018 even doubled the previous record year of 2006 in drained area and 10-fold in number of drained lakes. From 1950 until 2006/2007 lake drainage and lake expansion rates on the northern SP were fairly stable. The strong influence of large lakes on drainage rates in 2018 confirmed previous findings (Jones et al., 2011). A recent study by Swanson (2019) identified the same exceptional event for the National Parks of northwestern Alaska, which partially overlaps with our study area.

The high level of degradation, apparent by a large fraction of drained basins of several generations (Jones et al., 2012; Regmi et al., 2012), shows the general susceptibility of the landscape to rapid thermokarst lake dynamics, including drainage, within the study region. This landscape underwent intense thermokarst development over the past millennia, with the onset of thermokarst development during the early Holocene (Wetterich et al., 2012; Farquharson et al., 2016; Lenz et al., 2016). Available data of historic lake drainage is sparse. Therefore, a comparison of recent drainage rates with long-term development is difficult due to a lack of consistent observations, in particular for the pre-remote-sensing period.

**Local context**

The BP stands out in particular with a strong increase in lake drainage events in 2018, including its largest lake, relative to the period of 1999 to 2014. However, as evidenced by newly forming and expanding ponds, which are dotting the landscape,



active thermokarst lake expansion prevailed during the preceding decades, but rarely triggered drainage events. A recent study
by Jones et al. (2020b) has found a significant expansion of beaver dam building activities on the Baldwin Peninsula. Beavers
strongly influence local hydrological regimes by damming up thermo-erosional valleys or drained lake basins, which leads to
pond and lake formation and could potentially factor into lake drainage dynamics.
On the SP lake drainage was concentrated on the coastal region, particularly the Cape Espenberg lowlands, which are also hot-
spots of previous lake drainage events of the past decades. The location of recently drained lakes follows patterns of strong
Holocone thermokarst activity (Lenz et al, 2016) in the same area, which is apparent in highly degraded surface morphology.

**Influencing factors**

The exceptional weather conditions in western Alaska are likely the main cause of the significant lake drainage events of
summer 2018. Abundant snowfall with the second highest cumulative snowfall on record created a thicker-than-usual
insulation layer for the ground, which kept cold winter temperatures from penetrating the ground. This situation in combination
with record high winter temperatures, often just below freezing, likely led to an unfavorable energy balance for the stability of
permafrost. Both snow cover (Stieglitz et al., 2003, Ling and Zhang, 2003; Osterkamp, 2007) and winter temperatures are
important factors for near surface permafrost conditions. The abundant early winter snowfall in October and November in our
study area further increased the already strong snow insulation effect (Ling and Zhang, 2003).
The severe combination of both negative influencing factors very likely restricted the refreezing of the active layer and thus
potentially caused a landscape-wide talik development between the active layer and permafrost in 2018. Thinning of lake ice
during the ice growth season, also due to increased snow depth and warmer winter temperatures, has been identified as a factor
responsible for the shift from bedfast ice to floating ice on shallow lakes in several regions and to formation of new taliks
underneath lakes that previously were underlain by permafrost on the Alaska North Slope (Surdu et al., 2014; Arp et al., 2016;
Engram et al., 2018). In addition, later freeze-up in the fall period leads to a longer exposure of the near-shore lake-bed to
water, which likely increases permafrost destabilization and talik formation or growth along shores. Reports from Fairbanks
in interior Alaska, with a similar pattern of mild and snow-rich winter weather conditions, show that at various sites the active
layer did not refreeze completely during winter 2017/2018 (Farquharson et al., 2019b). The recent movement of beavers from
the treeline to tundra regions in northwestern Alaska could also be a contributing lake drainage mechanism that requires further
attention (Tape et al., 2018; Jones et al, 2020b).

**Temporal sequence and causes**

The occurrence of lake drainage around (during or shortly after) ice break-up indicates drainage driven by bank overtopping
or breaching in combination with rapid thermo-erosion during outflow. High water levels due to high precipitation in fall and
winter, rapid melt of abundant snow, in combination with destabilized lake margins, and possible talik formation have very
likely led to bank overflow or breaching of lake shores, and subsequent thermo-erosion and deepening of outflow channels.
The location of lakes within older drained lake basins with comparably unstable peaty and fine-grained substrates with high





intra-sedimentary ground ice contents as well as ice wedge networks enhanced the susceptibility of lakes to erosion and
drainage in addition to the weather induced driver.
Under current weather/climatic conditions with a MAAT around 0 °C, 5 °C above normal (1981-2010), permafrost aggradation
in the freshly exposed lake-beds might be slowed or even prevented, with consequences for basin hydrology and
biogeochemical cycling. After lake drainage, the lake-bed typically refreezes and permafrost soils can redevelop, which locks
in carbon stored in lacustrine sediments and terrestrial peat (Walter Anthony et al., 2014). *In-situ* measurements and continued
observations are necessary to test this hypothesis and determine whether this is happening already now on the SP and BP.

**Spatial comparison and considerations**

Western Alaska has been previously identified as one of the regions with the most intensive lake dynamics on a decadal-scale
(Nitze et al., 2017; Nitze et al., 2018a; Jones, et al.; 2011; Swanson, 2019; Jones et al., 2020b). Other regions along the
boundary of continuous permafrost in interior Alaska (Chen et al., 2014; Roach et al., 2013; Cooley et al., 2019) or the southern
Yamal Peninsula or western Siberia (Nitze et al., 2018a; Smith et al., 2003) are also highly affected by strong lake dynamics,
too, most notably lake drainage. Lake drainage is a common process in continuous permafrost of colder climates such as the
Arctic coastal plain of Alaska (Hinkel et al., 2007; Nitze et al., 2017; Jones et al., 2020a), Tuktoyaktuk Peninsula (Plug et al.,
2008; Olthof et al., 2015), Old Crow Flats (Labrecque et al., 2009; Lantz and Turner, 2015) or the Kolyma lowlands (Nitze et
al., 2017). However, lake dynamics tend to be of higher magnitude in warmer permafrost regions (Nitze et al., 2018a). In this
context, the drainage event of summer 2018 in our study region in western Alaska exceeded the average extent of lake area
loss by a factor of 7.5 and the previously most extreme year by 2.

**Data quality discussion**

The application of different methods and sensors, different temporal scales and varying spatial resolutions (long-term Landsat
datasets vs. Sentinel-1 water masks vs. Planet multi-temporal water masks) may introduce minor differences in masking water
and the delineation of water bodies. In a long-tailed distribution, as observed here, the widely used threshold of >25 % lake
area loss, strongly influences the number of drained lakes. For example, a threshold of >20 % lake area loss leads to an increase
from 192 to 279 drained lakes. However, the influence of total lake area loss remains low.
Due to the presence of lake-ice, the automated intra-annual lake tracking algorithm did not detect the early drainage events
reliably, however, the integration of multi-annual data into one analysis will highly benefit the automated lake tracking. With
the exponential growth of available data due to new satellite constellations (Sentinel-1, Sentinel-2, Planet), processing
platforms, and techniques, more reliable, better comparable, and spatially more extensive lake extent datasets will likely
become available in the near future.



**Outlook**

Extreme weather conditions of the winter year 2018 in western Alaska were driven by massively reduced sea ice cover in the Bering and Chukchi seas, resulting in much warmer and moister weather conditions than usual, which may have caused a so far unprecedented spatial and temporal clustering of lake drainage event in our study region. As climate models all predict a significant increase in both mean annual air temperature and precipitation for northern and western Alaska, the dramatic lake dynamics described here provide an early glimpse of the potentially massive changes in hydrology, permafrost, and topography to be expected in a warmer Arctic in similarly ice-rich permafrost landscapes. With MAAT around 0 °C, the years 2017 to 2019 already matched the MAAT projected for this region in ~2060 (RCP8.5) to beyond 2100 (RCP4.5) and precipitation projections for ~2080 (RCP8.5). This mismatch indicates that local to regional permafrost landscapes may experience much more severe and earlier impacts in a warming Arctic than what climate models are capable of predicting at fine scales. Permafrost degradation in northern Canada shows that drastic changes in the Arctic climate system can lead to processes which were projected to happen several decades later (Farquharson et al., 2019a).

The recent events potentially show the fate of lake-rich landscapes in continuous permafrost along its current southern margins, where near-surface permafrost degradation accelerates and permafrost will become discontinuous in the next decades. The colder less dynamic lake-rich coastal plain of northern Alaska may become more dynamic once climatic patterns will have moved towards the middle-to-end of the century.

**6 Conclusion**

The lake-rich northern Seward and Baldwin peninsulas in northwestern Alaska were affected by unprecedented lake drainage in 2018, which dwarfed previous lake changes of this historically dynamic permafrost landscape. Due to the mean annual air temperatures of this region reaching values close to 0 °C in combination with exceptional precipitation in recent years, matching model projections for the years 2060 (RCP8.5) to 2100 (RCP4.5), near-surface permafrost is likely already in a phase of degradation and destabilization around the lake margins. This in combination with rapid availability of excess surface water likely caused the rapid drainage of nearly 200 lakes during or shortly after ice-break up in 2018, including some of the largest lakes of the region that likely persisted for several millennia. Under a rapidly warming and wetting climate, in conjunction with ongoing sea ice loss in the Bering Strait, we expect a further intensification of permafrost degradation, reshaping the landscape and a transition from continuous to discontinuous permafrost, and significant changes in hydrology and ecology. The impact on habitat and landscape characteristics will be drastic in these formerly lake-rich regions. The recent processes observed in northwestern Alaska potentially will be a precedent for lake dynamics of rapidly warming lake-rich permafrost landscapes approaching the MAAT threshold of 0 °C.





**Competing Interests**
The authors declare that there are no competing interests.
**Code and Data**
**Data**
Supplementary figures and tables data can be found in the supplementary file.
**Lake datasets:**
https://github.com/initze/NW_Alaska_Drainage_Paper/tree/master/figures/lake_datasets

**Intra-annual lake area plots:**
https://github.com/initze/NW_Alaska_Drainage_Paper/tree/master/figures/lake_drainage/planet_lake_area

**Weather and climate plots:**
https://github.com/initze/NW_Alaska_Drainage_Paper/tree/master/figures/weather_and_climate/

**Lake drainage animations:**
https://github.com/initze/NW_Alaska_Drainage_Paper/animations/lake_animations_drainage_1999-2014/

Final lake change datasets will be published on the PANGAEA data repository

**Code**
**Sentinel-1 Watermasks Google Earthengine Script:**
https://code.earthengine.google.com/7d2367758eead1614202efcfa6bed2b5
**Landsat Video Animation Google Earthengine Script:**
https://code.earthengine.google.com/c879add607322305b8293904bea6d781
**noaaplotter weather plotting package:**
https://github.com/initze/noaaplotter



## Acknowledgements

IN and GG were supported by ERC PETA-CARB (#338335), ESA GlobPermafrost, ESA CCI+ Permafrost, and HGF AI-CORE. BMJ was supported by the National Science Foundation awards OPP-1806213. SC was supported by a National Science Foundation Graduate Research Fellowship. CD was supported by the Natural Sciences and Engineering Research Council of Canada (NSERC) grant number RGPIN-05049-2017.

We acknowledge help during field work by J. Lenz, M. Fuchs, and J. Strauss.

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

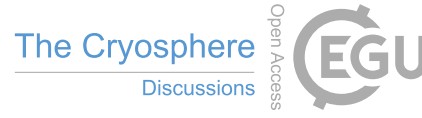


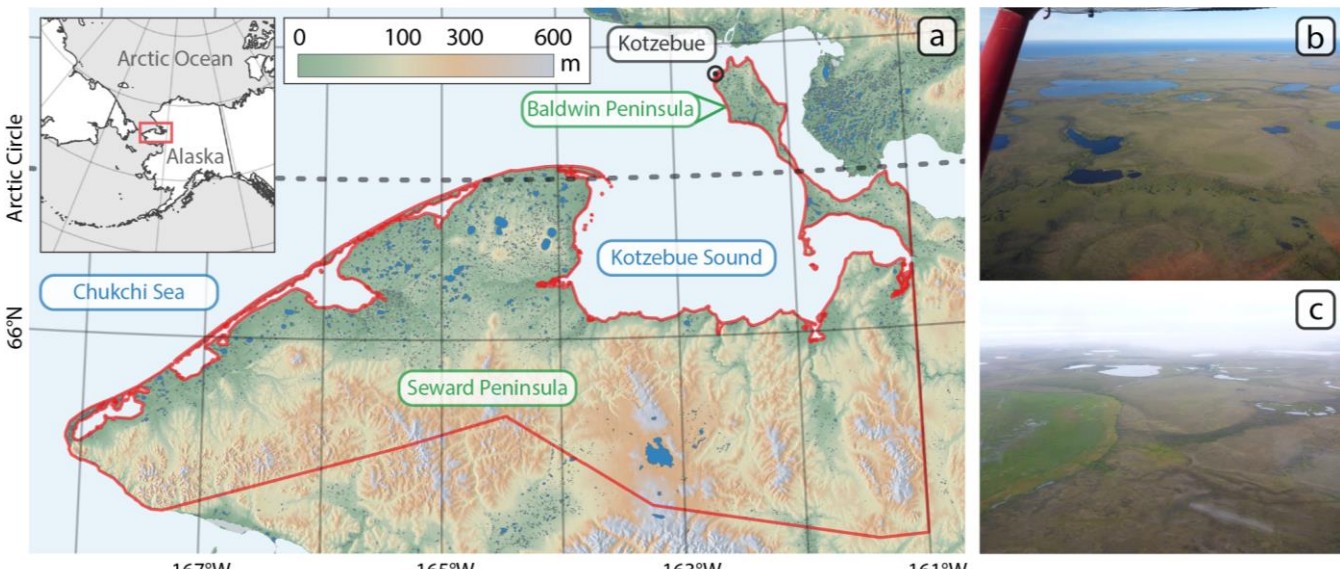


**Figure 1: a) Overview of study area with topography and place names. Elevation source: GMTED2010. b) Oblique aerial photo of the formerly largest lake on the Baldwin Peninsula, which drained in 2018. Photo: J.Strauss, July 2016. c) Oblique aerial photo of the northern Seward Peninsula. Photo: G.Grosse, July 2016. Lake-rich permafrost landscape with large drained basin.**







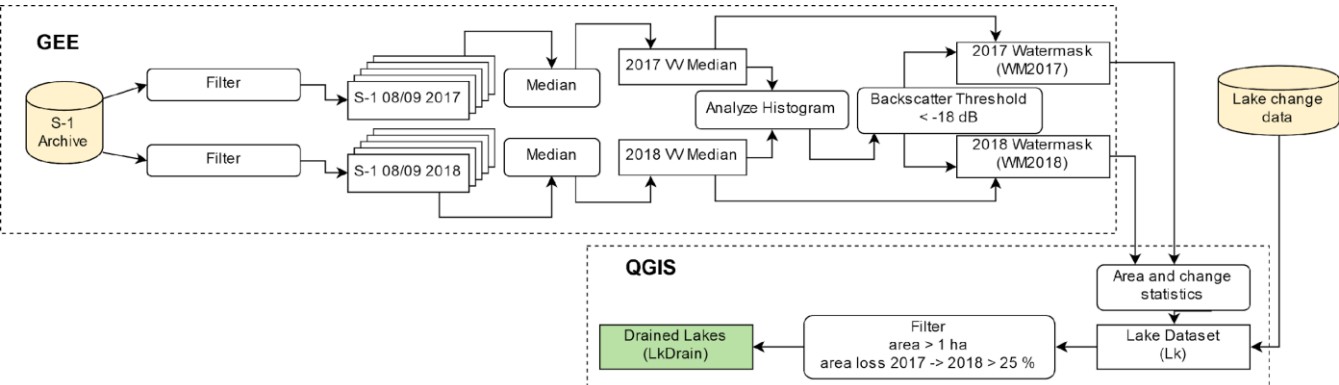


**Figure 2: Flowchart of lake change detection and drainage assignment based on Sentinel-1 data (S-1 Archive). Raster data processing was carried out in Google Earthengine (GEE). Lake vector extraction and calculation of recent and historic (Lake change data: Nitze et al., 2018b) lake change statistics was carried out in Quantum GIS (QGIS).**





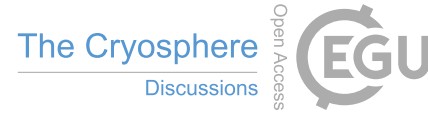


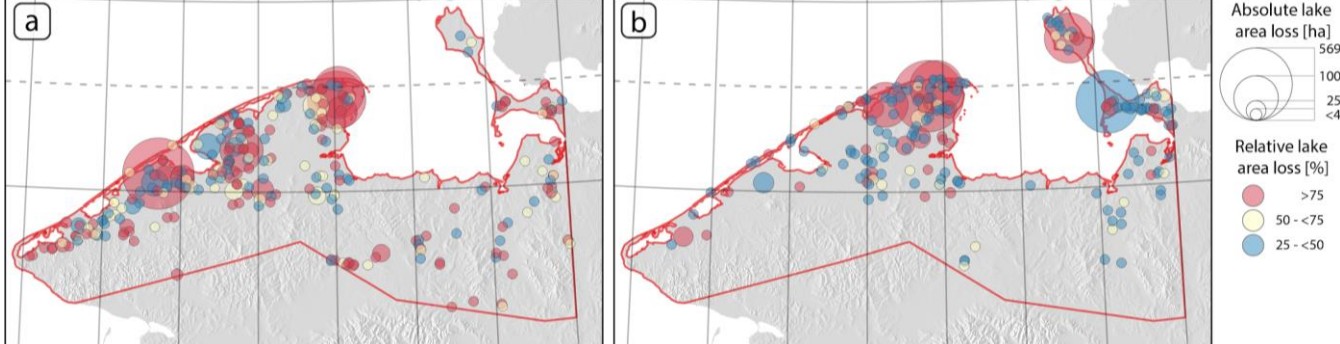


**Figure 3: Spatial patterns, size and percentage of drained lakes. a) 1999-2014, b) 2017-2018. Hillshade based on the GMTED2010 elevation dataset.**







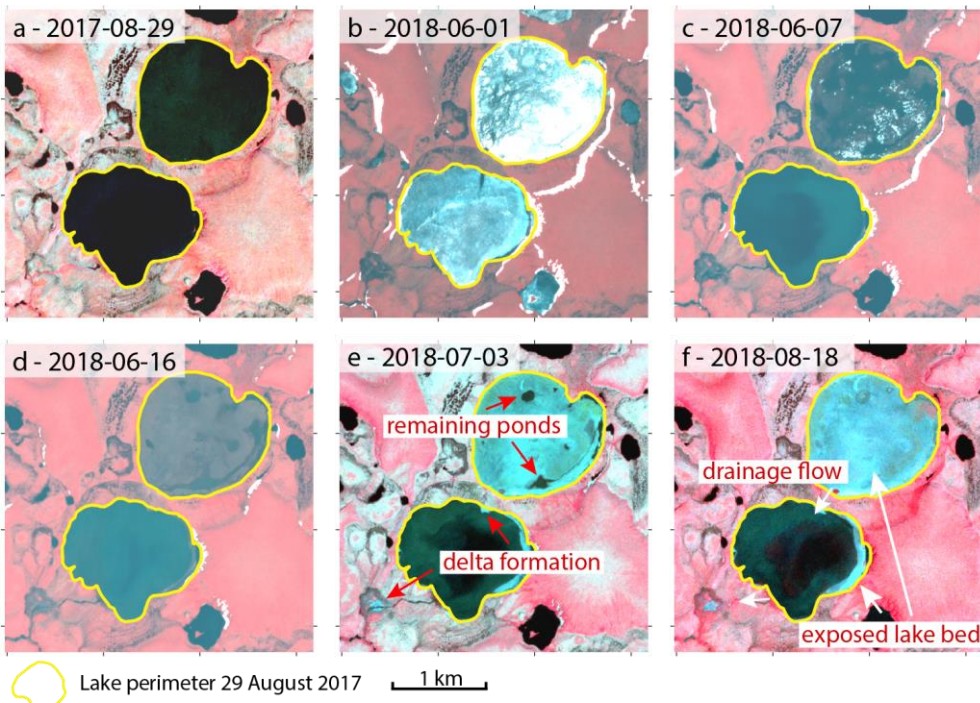


**Figure 4: PlanetScope (Planet Team, 2017) satellite time-series of cascaded lake drainage of lakes 99492 (north) and 99522 (south) (66.45°N, 164.75°W) from 29 August 2017 until 18 August 2018 with annotations of drainage related features. a) Lakes before drainage. b) Ice-break-up with initial drainage pattern visible on the northern lake. c) Post ice-breakup with reduced water level in the northern lake. d) Northern lake nearly completely drained with few remaining ponds. e) Partial drainage of the southern lake with visible delta formation. f) FInal stage of lake drainge with dried out ponds (northern lake) and lake level stabilization (southern lake).**





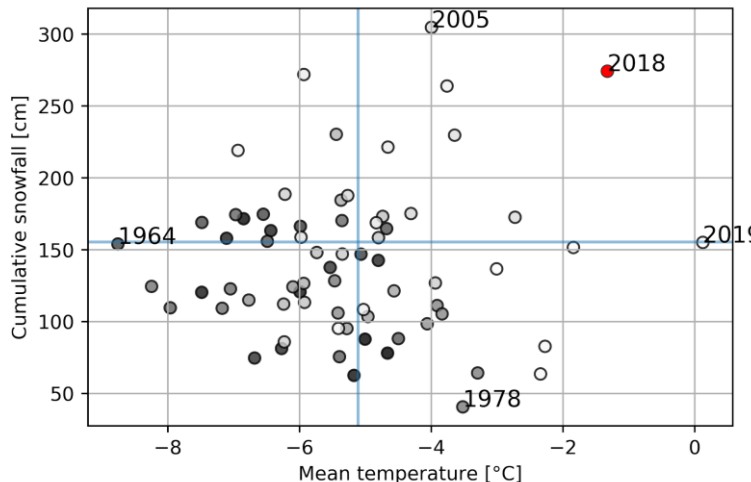

**Figure 5: Scatterplot of mean air temperature and cumulative snowfall per winter year (July to June). Winter year 2017/2018 marked in red. Extreme years indicated by number. Blue lines indicate climatic means of MAAT and cumulative snowfall (1981-2010). Dots in greyscale indicate the year from 1934 (black) until 2019 (white).**





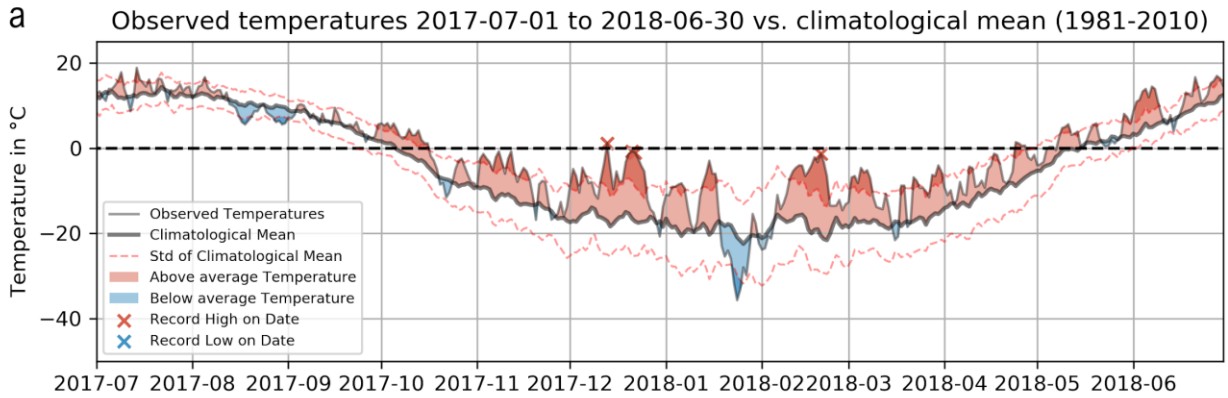

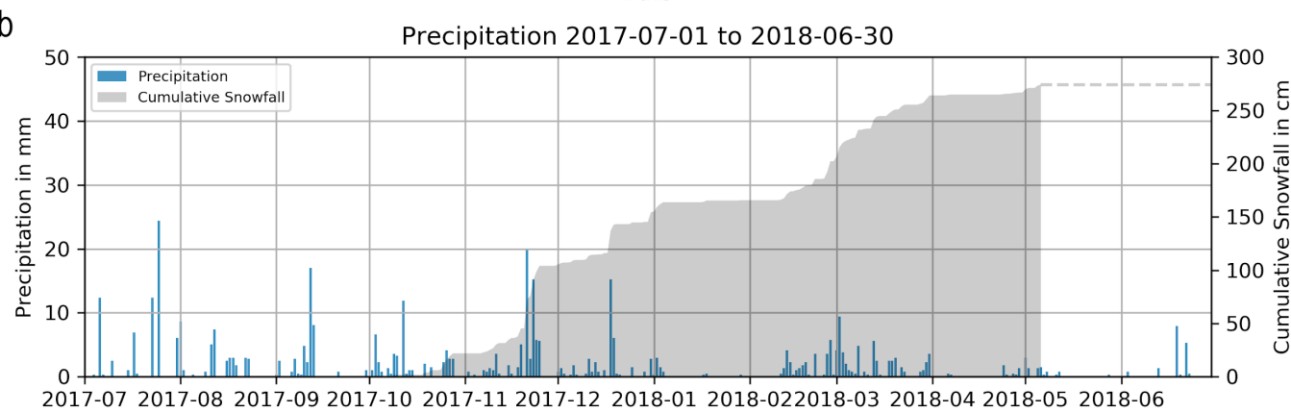


Figure 6: Overview of winter weather conditions at Kotzebue climate station from July 1 2017 through June 30 2018. a) Observed
temperatures in °C with anomaly (red: warmer, blue: colder) from climatological mean (1981-2010). Dark color shades indicate
deviation of >1 standard deviation from the mean. Record temperatures for particular days are marked with an "x" b) Daily
precipitation and cumulative snowfall.





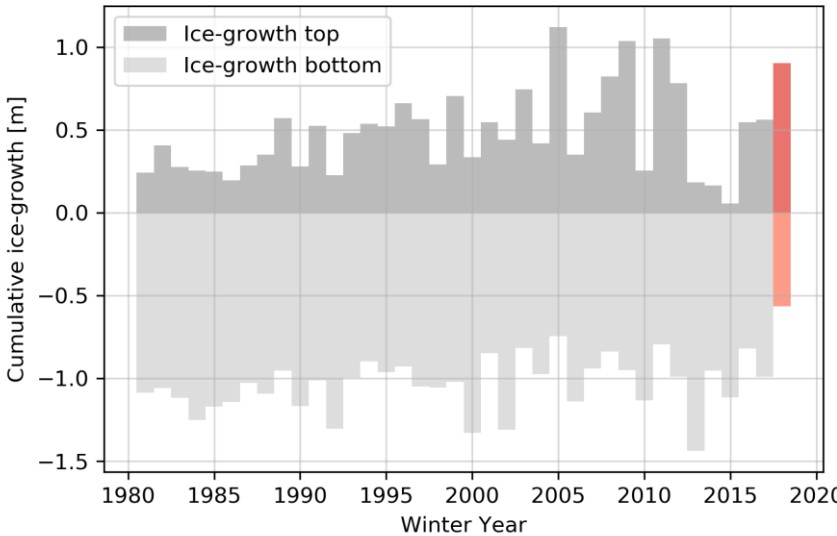


**Figure 7: Simulated cumulative top and bottom ice-growth per winter year for 100 % snow scenario in cm. Winter year 2017/2018 highlighted in red.**







**Table 1: Overview of datasets and for lake change analysis.**

| Dataset | Abbreviation | Period | Source |
|---|---|---|---|
| Lake Change Dataset | Lk | 1999-2014 | Nitze et al. 2018b |
| Watermask Sentinel 1 2017 | WM 2017 | 2017 | |
| Watermask Sentinel 1 2018 | WM 2018 | 2018 | |
| Planet dynamic water mask | LkDyn | 2017-2018 | |
| Derived Lake change 2017-2018 | LkDrain | 1999-2018 | |


**Table 2: Lakes ranked by largest area loss from 2017 to 2018 with lake area rank 2017, Lake ID, net change area and percentage as well as lake area in 2017 and 2018. For full dataset see Supplementary Table 1 and datasets LkDrain. *Lagoon connected to the sea/Kotzebue Sound.**

| Drain rank | Lake area rank 2017 | Lake ID | Net change 2017-2018 [ha] | Net change 2017-2018 [%] | Area 2017 [ha] | Area 2018 [ha] | Video animation |
|---|---|---|---|---|---|---|---|
| 1 | 12 | 99368 | -332.04 | -91.24 | 363.92 | 31.88 | Link |
| 2* | 6 | 69152 | -258.8 | -34.31 | 754.36 | 495.56 | Link |
| 3 | 32 | 99230 | -185.12 | -99.70 | 185.68 | 0.56 | Link |
| 4 | 39 | 64656 | -164.6 | -99.83 | 164.88 | 0.28 | Link |
| 5 | 51 | 99492 | -132.12 | -100 | 132.12 | 0 | Link |
| 6 | 205 | 100218 | -28.48 | -78.24 | 36.4 | 7.92 | Link |
| 7 | 105 | 101659 | -27.56 | -41.53 | 66.36 | 38.8 | Link |
| 8 | 269 | 99545 | -26.12 | -97.32 | 26.84 | 0.72 | Link |
| 9 | 281 | 102499 | -25.72 | -100 | 25.72 | 0 | Link |
| 10 | 305 | 100470 | -23.2 | -100 | 23.2 | 0 | Link |








**Table 3: Lakes with largest area loss from 1999 to 2014 with net change area and percentage as well as lake area of 1999 and 2014 (Nitze et al., 2018b). For full datasets see Supplementary Table 2 and datasets Lk.**

| Drain Rank | Lake ID | Net change 1999-2014 [ha] | Net change 1999-2014 [%] | Area 1999 [ha] | Area 2014 [ha] | Year Drained |
|---|---|---|---|---|---|---|
| 1 | 101282 | -568.92 | -97.95 | 580.8 | 11.88 | 2007 |
| 2 | 99433 | -373.29 | -99.63 | 374.67 | 1.37 | 2006 |
| 3 | 99313 | -299.98 | -78.77 | 380.84 | 80.85 | 2006 |
| 4 | 100588 | -208.53 | -94.69 | 220.22 | 11.69 | 2004 |
| 5 | 99624 | -113.43 | -99.55 | 113.94 | 0.52 | 2006 |
| 6 | 101659 | -79.32 | -31.55 | 251.42 | 172.1 | 2009 |
| 7 | 99505 | -76.16 | -62.7 | 121.48 | 45.31 | 2003 |
| 8 | 100505 | -74.27 | -28.86 | 257.36 | 183.08 | 2003 |
| 9 | 101402 | -65.62 | -98.3 | 66.75 | 1.14 | 2003 |
| 10 | 101844 | -56.5 | -99.06 | 57.03 | 0.54 | 2004 |








**Table 4. Annually aggregated observations of mean air temperature, cumulative precipitation, cumulative snowfall, cumulative**
**freezing degree days, and freezing days per winter year (July 1 until June 30) for climate station Kotzebue, sorted by mean air**
**temperature. 10 warmest and 5 coldest years included. For full data (1950-2019) see Supplementary Table 3.**

| Winter Year | Rank Temperature | Mean Air Temperature [°C] | Cumulative Precipitation [mm] | Cumulative Snowfall [cm] | Cumulative FDD | Freezing Days |
|---|---|---|---|---|---|---|
| 2019 | 1 | +0.12 | 278.6 | 155.1 | -1755.50 | 181 |
| 2018 | 2 | -1.33 | 424.5 | 274.2 | -1904.75 | 196 |
| 2016 | 3 | -1.84 | 258.1 | 151.6 | -2142.85 | 200 |
| 2014 | 4 | -2.27 | 260.5 | 82.8 | -2136.75 | 178 |
| 2015 | 5 | -2.34 | 247.8 | 63.6 | -2428.80 | 208 |
| 2003 | 6 | -2.73 | 244.0 | 172.6 | -2262.85 | 195 |
| 2017 | 7 | -3.01 | 225.0 | 136.7 | -2631.05 | 194 |
| 1979 | 8 | -3.29 | 207.5 | 64.3 | -2648.80 | 221 |
| 1978 | 9 | -3.52 | 210.9 | 40.7 | -2795.10 | 206 |
| 2004 | 10 | -3.64 | 313.5 | 229.7 | -2698.15 | 181 |
|  |  |  |  |  |  |  |
| 1966 | 64 | -7.48 | 262.7 | 169.0 | -3642.30 | 240 |
| 1955 | 65 | -7.48 | 305.9 | 120.4 | -3711.65 | 225 |
| 1971 | 66 | -7.96 | 160.3 | 109.6 | -3975.45 | 237 |
| 1976 | 67 | -8.25 | 199.7 | 124.5 | -3923.10 | 239 |
| 1964 | 68 | -8.76 | 300.6 | 154.0 | -4130.00 | 227 |

