# Peer review of "The catastrophic thermokarst lake drainage events of 2018 in"

_The Cryosphere, 2020_

## Referee Comment (RC1) · D.K. Swanson (Referee) · 18 Jun 2020

This is an excellent paper. It makes a significant contribution to our understanding of this alarming new phenomenon of thermokarst lake drainage. I was particularly interested to see the detailed analysis of the seasonal timing of the drainage events and the implications for the processes involved. Below are some minor comments, identified by line number in the version "tc-2020-106-manuscript-version2.pdf"

l 99-101. Be sure to specify that these are "mean annual ground temperatures". Part of the study area is covered by models of Panda et al 2016. Panda, Santosh K., Sergei S. Marchenko, and Vladimir E. Romanovsky. 2016. "High-Resolution Permafrost Modeling in the Arctic Network of National Parks, Preserves and Monuments." Natural Resource Report NPS/ARCN/NRR-2016/1366. Fort Collins, Colorado: National Park Service. https://irma.nps.gov/DataStore/Reference/Profile/2237720.

They show a little colder temperatures, for 2000-2009 anyway. You should mention what the time interval is for the Obu temperatures, since temperatures are changing so fast.

l 106, 110. Not sure of the meaning of "strongly degraded" and "highly degraded" here. There are many "healthy" low-center ice-wedge polygons here, and drained lake basins of many different ages, suggesting an ongoing process. There certainly is a lot of thermokarst, but until recently the area had continuous permafrost that was not degrading. Former yedoma in this area presumably degraded in many places, is that what you mean? This probably happened long ago, perhaps in the early Holocene for the most part. I consider this area to be a typical thaw-lake plain, with healthy permafrost until recently.

l 158. So some lakes probably drained between your 2014 and 2017 data sets. It looks like you didn't try to quantify what went on 2014-2017, is this because you were concerned about the differences in water recognition between Landsat (1999-2014) and SAR (2017-2018)? A critical reader might ask for evidence that there wasn't a big drainage year during this interval too. To answer this possible objection you could do an approximate count/area of lakes that drained 2014-2017, based on your data. I'm pretty sure it will show that not many lakes drained then, which is all you really need to say.

l 212. I understand from talking to co-workers that the authors are already aware of the erroneously high temperature values that were posted for Kotzebue in 2019, and that may have found there way into these summaries.

l 258-259. As topographic lows I would expect small lakes, and shores of large lakes, to have drifts with substantially more than the weather station measured snow depth.

l 284-285. Its not clear if there were 9 pre-July drainage events and 1 July, or 8 pre-July drainages and 1 July.

l 309. These are estuaries with salt marshes that are periodically flooded. Presumably you excluded other lagoons and estuaries from the study area, you could exclude this one. The processes controlling flooding/drainage here are different from the thermokarst lakes. Marine sand deposition blocks the mouth of the estuary. Flooding could be caused by this blockage of river runoff, or by storm tides. It looks like there is also thermokarst on the upstream end of the estuary, where they isn't as much marine influence. But in any case it isn't a permafrost-thaw process that causes the water level here to go up and down.

l 348. I understand that the June 2019 temperatures at Kotzebue were erroneously high, so this mean annual average of 0.12 deg C might be exaggerated slightly. But the overall story of temperatures rising to near freezing should not be affected.

l 350-351. Its not clear which year these numbers refer to.

l 352. Table 4 not 5.

l 362-363. The use of "increase to" and "increase of" is grammatically correct but a little confusing here. Also, the increase of 3.7 to 6.6 C is relative to some older average, you shouls say what the older time period was.

l 403. Another reference to degraded surface morphology. Does this mean pervasive thermokarst?

l 421-422. There's no evidence of beavers in the Espenberg region that I know of. I don't think they had any role in any of the big northern Seward Peninsula lake drainages that you describe. In the Kobuk valley I've seen where they dam up the outlet of a drained lake, allowing it to refill.

l 431. "drainage, in addition to the weather-induced driver."

l 464. "so-far"

l 475-477. I expect that the North Slope will see a similar outbreak of lake drainages when its temperatures hit 0 C also. Do you have any predictions about if and when that will occur?

l 483-484. "This in combination with excess surface water likely caused the rapid drainage ...". This sentence is also a bit long/run-on.

---

## Referee Comment (RC2) · Anonymous Referee #2 · 20 Jul 2020

This paper presents interesting and novel research on how quickly lake drainage can happen over a substantial area when extreme weather events occurs. The paper is overall good and easy to follow but there are a few comments that I would suggest that the authors consider before publication.

Major comments: - Why did you choose to do the comparison between 1999-2014 and 2017-2018 and leave out 2015 and 2016? Would be good if you could motivate this as I assume it has a scientific reason. - In the aim it is stated that this study should investigate weather and climate data as well as modelled lake ice conditions as potential drivers of the widespread lake drainage. The results from the modelled

part are not well covered in the discussion. At present, section 4.2.2. and 4.3 can be deleted or the results should be better incorporated in the discussions. - Why did you choose to work on lakes larger than 1 ha? I assume you could have included smaller lakes as well with the resolution of your data set and given the possible importance of the smaller lakes for GHG emissions (See e.g. Kuhn et al., 2018. Emissions from thaw ponds largely offset the carbon sink of northern permafrost wetlands. Scientific reports), it would be great if you could please add a sentence about why you chose to only work on lakes with this size. - In the discussion the influencing factors are discussed, many sentences states that it is likely.... Would it not be possible to make a multiple regression with the climate parameters (that have already been analysed) to see if you have any statistically significant connections?

Minor comments: Line 34 – Brown et al., 1997 is missing from the ref list Line 39 – Nitze et al., 2018 should it be a or b? Line 40 – Pastick et al., 2015 is missing from the ref list Line 41 – Liljedahl et al., 2015 is missing from the ref list Line 53 – Jones and Arp, 2015 is missing from the ref list Line 65 – Lawrence and Slater, 2005 is missing from the ref list Line 70 – Nitze et al.. 2018 should it be a or b? Line 71 – Nitze et al.. 2018 should it be a or b? Line 109 – Hopkins et al., 1955 should be Hopkins, 1955? Line 188 – Nitze et al., 2018 should it be a or b? Line 204-205 – Lakes where the timing could not be detected manually, e.g. in case of very subtle drainage, were assigned no drainage year (25 of 270); what does the numbers in the parentheses mean? Line 212 – Perhaps a good idea to refer to Figure 1a after Kotzebue? Line 281 – I suggest to remove the heading 4.1.2 as then you will have the same style for both 2017-2018 comparison and the past comparison. Line 352 – refers to Table 5 which I could not find in the manuscript Line 435 - Walter Anthony et al., 2014 is not in the reference list Line 442 – Smith et al., 2003 is not in the reference list Line 443 – remove too in the beginning of this line Line 642 - Nitze and Grosse, 2016, is this reference referred to in the running text? Line 660 – Pastick et al., 2019, is this reference referred to in the running text? Line 713 – Figure 1. This figure is fairly "empty". Consider to include the lakes from table 2 in the map. Maybe also include Nome that you mentions weather

data from in section 4.2.1? Line 727 – Figure 3. What does hillshade stand for? I cannot detect it in the figure Line 732 – Figure 4 – Suggest to move "remaining pools" to figure d where it is mentioned. Line 739 – Figure 5 – I think it is quite confusing with the greyscale on the dots, maybe you can have one colour per decade or something (with a legend outside of the box).

---

## Author Comment (AC1) · 10 Aug 2020

l 348. I understand that the June 2019 temperatures at Kotzebue were erroneously high, so this mean annual average of 0.12 deg C might be exaggerated slightly. But the overall story of temperatures rising to near freezing should not be affected.

We acquired the latest NOAA data on 29 July 2020. In this new updated dataset temperature data from 1 May through 3 September 2019 indeed were removed from NOAA due to erroneous measurements. For winter year 2019 we calculated the temperature difference during the period with available data, 1 July through 30 April, and interpolated until 30 June 2019. We stated in the text and captions that these values are

interpolated. We updated sections 3.3.1 and 4.2.1. We furthermore updated Figure 5, Table 4, and Supplementary Table 3 and marked 2019 with an asterisk (*). Precipitation and snowfall data remained unaffected by the weather station issues during 1 May - 3 September 2019. Reviewer 1 is correct that the overall story of temperatures rising to near freezing is still valid.

l 350-351. Its not clear which year these numbers refer to.

It now reads "During winter year 2018 the weather station Kotzebue recorded only ..."

l 352. Table 4 not 5.

Changed to "Table 4"

l 362-363. The use of "increase to" and "increase of" is grammatically correct but a little confusing here. Also, the increase of 3.7 to 6.6 C is relative to some older average, you shouls say what the older time period was.

We changed the sentence to "... which marks an increase of 3.7 to 6.6 °C compared to the period from 2010-2019 ..."

l 403. Another reference to degraded surface morphology. Does this mean pervasive thermokarst?

We changed the structure of the sentence and added: "The highly degraded surface morphology in this region indicates active and pervasive thermokarst processes."

l 421-422. There's no evidence of beavers in the Espenberg region that I know of. I don't think they had any role in any of the big northern Seward Peninsula lake drainages that you describe. In the Kobuk valley I've seen where they dam up the outlet of a drained lake, allowing it to refill.

Over the past two decades the beaver population has expanded significantly across NW Alaska, which is shown in Tape et al., 2018 and Jones et al. 2020. The latter found strong beaver activity on the Baldwin Peninsula, which is part of the study area.

Although currently, beaver activity is unlikely at the (coastal) Cape Espenberg region, beaver dams were detected on the southern part of the northern Seward Peninsula with recent very-high-resolution satellite imagery. We changed the wording of the sentence to focus more on lake dynamics, as beavers are at the moment more responsible for lake growth: "The recent movement of beavers from the treeline to tundra regions in northwestern Alaska could also be a contributing driver of lake dynamics in the eastern and southern portions of the study region that requires further attention (Tape et al., 2018; Jones et al, 2020b)."

l 431. "drainage, in addition to the weather-induced driver."

We split the sentence into two separate sentences for easier readability.

l 464. "so-far"

We removed "so far"

l 475-477. I expect that the North Slope will see a similar outbreak of lake drainages when its temperatures hit 0 C also. Do you have any predictions about if and when that will occur?

The SNAP model ensemble for scenario RCP8.5 predicts MAAT of -2.6±0.3 °C and MAP of 315 mm for the southern (lake-rich) part of the Arctic Coastal Plain. However, extreme events with high temperatures and precipitation may be likely much earlier, as seen in W/NW Alaska during 2017-2019. We added the following sentence: "Temperatures are predicted to approach 0°C (MAAT 2090-2099: -2.6°C) on the southern Arctic Coastal Plain in a RCP8.5 climate scenario."

l 483-484. "This in combination with excess surface water likely caused the rapid drainage ...". This sentence is also a bit long/run-on.

We split this sentence for better readability.

---

## Author Comment (AC2) · 10 Aug 2020

This paper presents interesting and novel research on how quickly lake drainage can happen over a substantial area when extreme weather events occurs. The paper is overall good and easy to follow but there are a few comments that I would suggest that the authors consider before publication.

Major comments: - Why did you choose to do the comparison between 1999-2014 and 2017-2018 and leave out 2015 and 2016? Would be good if you could motivate this as I assume it has a scientific reason.

The two comparison periods are based on data availability. The Planet data has only been available in high temporal resolution for our study region since 2017/2018. As a base for comparison, the 1999-2014 lake extent layer originally produced by Nitze et al 2018a and 2018b was readily available for our analysis. We added a sentence in 3.1.1 to clarify that the 1999-2014 dataset is readily available. We furthermore added a sentence in 3.1.2 why we focused on the change from 2017 to 2018 in the SAR data analysis.

In the aim it is stated that this study should investigate weather and climate data as well as modelled lake ice conditions as potential drivers of the widespread lake drainage. The results from the modelled part are not well covered in the discussion. At present, section 4.2.2. and 4.3 can be deleted or the results should be better incorporated in the discussions

We expanded the discussion with a more thorough analysis and discussion of the lake ice model results.

Why did you choose to work on lakes larger than 1 ha? I assume you could have included smaller lakes as well with the resolution of your data set and given the possible importance of the smaller lakes for GHG emissions (See e.g. Kuhn et al., 2018. Emissions from thaw ponds largely offset the carbon sink of northern permafrost wetlands. Scientific reports), it would be great if you could please add a sentence about why you chose to only work on lakes with this size. –

The data analysis from Nitze et al., 2018a (Dataset: Nitze et al, 2018b) was based on Landsat data with 30m spatial resolution. The minimum mapping unit was set to 1 ha to avoid excessive uncertainties. Therefore, by using this dataset we use the same minimum mapping unit. We added the Sentence "Water bodies smaller than 1 ha were excluded from the original analysis due to Landsat's spatial resolution of 30m." to clarify that we use the minimum mapping unit of the original dataset.

In the discussion the influencing factors are discussed, many sentences states that it is

likely.... Would it not be possible to make a multiple regression with the climate parameters (that have already been analysed) to see if you have any statistically significant connections?

We carried out a multivariate RandomForest (Breiman, 2001) regression with annual weather attributes as input features and drained lake area per year (1999-2014, 2018) as the target variable. We used the Random Forest internal Feature Importance metric, which is widely used to quantify an input variable's importance. We furthermore evaluated the model performance during training and independent validation using $r^2$. We added a new subsection to the methods (3.5), and results (4.5) sections. We expanded the discussion with the results.

Minor comments:

Line 34 – Brown et al., 1997 is missing from the ref list

The reference has been added.

Line 39 –Nitze et al., 2018 should it be a or b?

We changed the reference to Nitze et al., 2018a

Line 40 – Pastick et al., 2015 is missing from theref list

The reference has been added.

Line 41 – Liljedahl et al., 2015 is missing from the ref list

The reference has been added and changed to Liljedahl et al., 2016.

Line 53 – Jones and Arp, 2015 is missing from the ref list

The reference has been added.

Line 65 – Lawrence and Slater, 2005 is missing from the ref list

The reference has been added.

Line 70 – Nitze et al.. 2018 should it be a or b?

We changed the reference to Nitze et al., 2018a

Line 71 – Nitze et al..2018 should it be a or b?

We changed the reference to Nitze et al., 2018a

Line 109 – Hopkins et al., 1955 should be Hopkins, 1955?

We changes the reference to Hopkins, 1955.

Line 188 – Nitze et al., 2018 should it be a or b?

We changed the reference to Nitze et al., 2018b

Line 204-205 – Lakes where the timing could not be detected manually, e.g. in case of very subtle drainage, were assigned no drainage year (25 of 270); what does the numbers in the parentheses mean?

This part now reads "(25 of 270 lakes)" to clarify that 25 of 270 lakes do not have a drainage year.

Line 212 – Perhaps a good idea to refer to Figure 1a after Kotzebue?

We added "(see Figure 1a)"

Line 281 – I suggest to remove the heading 4.1.2 as then you will have the same style for both 2017-2018 comparison and the past comparison.

We adapted the numbering of sections 3 (Methods) and 4 (Results) that subsection numbering matches

Line 352 – refers to Table 5 which I could not find in the manuscript

Changed to "Table 4"

Line 435 - Walter Anthony et al., 2014 is not in the reference list

The reference has been added.

Line 442 – Smith et al., 2003 is not in the reference list

We changed the reference to Smith et al., 2005

Line 443 – remove too in the beginning of this line

"too" is now removed

Line 642 - Nitze and Grosse, 2016, is this reference referred to in the running text?

The reference is now removed

Line 660 – Pastick et al., 2019, is this reference referred to in the running text?

The reference is now removed

Line 713 – Figure 1. This figure is fairly "empty". Consider to include the lakes from table 2 in the map. Maybe also include Nome that you mentions weather data from in section 4.2.1?

We would like to keep the "empty" design as the map already contains several information layers (land/water, elevation, large lakes, place names) in our opinion as this map is targeted to provide a general overview of the region. Results (including large lake drainage events) are shown in Figure 3, which focusses on the results (with fewer information layers). We have added a reference to Nome on the bottom of the map.

Line 727 – Figure 3. What does hillshade stand for? I cannot detect it in the figure

The hillshade is a shaded relief map, based on a digital elevation model, which we used here as a visual background of the land area. We changed the wording of the caption accordingly.

Line 732 – Figure 4 – Suggest to move "remaining pools" to figure d where it is mentioned.

We moved the annotations "remaining pools" to Figure 4d. Furthermore we improved the color scaling for better contrast between different landscape features.

Line 739 – Figure 5 – I think it is quite confusing with the greyscale on the dots, maybe you can have one colour per decade or something (with a legend outside of the box).

We changed the color coding to decadal steps for better visual separation. A legend is attached.